# *Neofusicoccum cryptomeriae* sp. nov. and *N. parvum* Cause Stem Basal Canker of *Cryptomeria japonica* in China

**DOI:** 10.3390/jof9040404

**Published:** 2023-03-24

**Authors:** Yuan-Zhi Si, Jian-Wei Sun, Yu Wan, Yi-Na Chen, Jiao He, Wei-Zheng Li, De-Wei Li, Li-Hua Zhu

**Affiliations:** 1College of Forestry, Nanjing Forestry University, Nanjing 210037, China; 2Co-Innovation Center for Sustainable Forestry in Southern China, Nanjing Forestry University, Nanjing 210037, China; 3Jiangsu PIESAT Information Technology Co., Ltd., Xuzhou 221116, China; 4Zhouning County Bureau of Forestry, Ningde 355400, China; 5Advanced Analysis and Testing Center, Nanjing Forestry University, Nanjing 210037, China; 6The Connecticut Agricultural Experiment Station Valley Laboratory, Windsor, CT 06095, USA

**Keywords:** *Cryptomeria japonica*, *Neofusicoccum*, multi-locus phylogeny, new disease

## Abstract

*Cryptomeria japonica* D. Don is a coniferous tree species widely grown in southern China for its high ornamental value. Recently, during disease surveys in China, a symptom of dieback occurred on *C. japonica* in Nanjing, Jiangsu Province, China. A total of 130 trees were surveyed and more than 90% showed the same symptom. The crowns of affected trees were brown when viewing from a distance, and the bark showed no difference from the healthy ones. In this study, 157 isolates were isolated from the 3 affected plants of *C. japonica*, and based on the living culture on PDA, the fungal isolates were preliminarily divided into 6 groups. Thirteen representative isolates were selected for the pathogenicity test, and seven of them showed obvious pathogenicity on *C. japonica,* causing stem basal canker. These isolates were identified based on comparisons of the DNA sequences of the internal transcribed spacer regions (ITS), partial translation elongation factor 1-alpha (*tef1*), β-tubulin (*tub2*), and DNA-directed RNA polymerase II subunit (*rpb2*) and combined with their morphological characteristics. Results showed that these seven isolates belong to two taxa in *Neofusicoccum*, including a species new to science. The new species, *Neofusicoccum cryptomeriae,* was hereby described and illustrated. The other species was *N. parvum*. Both species were pathogens of stem basal canker of *Cryptomeria japonica*.

## 1. Introduction

*Cryptomeria* is a monotypic genus of conifer in *Cupressaceae*, and its only species, *Cryptomeria japonica* (Linn. f.) D. Don, Japanese cedar, is a monoecious coniferous tree species native to Japan and has been introduced to Jiangsu, Guangxi, Shandong, Zhejiang, and other provinces in China as an ornamental tree species and for lumber production [1,2]. *Cryptomeria japonica* has been introduced and cultivated in China for a millennium or more, and some trees on Tianmu Mountain are estimated to be nearly 1000 years old, and the oldest one, ca. 1500 years old [3,4]. It has been introduced to other countries: Azores (Portugal), former Czechoslovakia, Føroyar (Denmark), Korea, Mauritius, New Zealand, Réunion (France), Sweden, Turkey, and the UK [2].

*Cryptomeria japonica* has extensive application values. Its cones and unusual needles have high aesthetic appeal. As an environmental-friendly tree, *C. japonica* showed a good effect on the absorption of cesium [5,6]. Many studies have shown that the bark, core material, and needles of *C. japonica* contain a variety of monoterpenoids, sesquiterpenes, and diterpenoids [7]. These substances show a wide range of biological activities, such as antibacterial and insect resistance [8,9,10], and the hepatoprotective phytocompounds from *C. japonica* have a potential function in inflammatory mediators [11]. In addition, as a common timber species, *C. japonica* was widely used in the production of particleboard [12]. However, during the seedling stage and the afforestation process, *C. japonica* was often attacked by many kinds of pathogenic fungi, causing a number of diseases, including the trunk rot caused by *Fomitiporia torreyae* Y. C. Dai & B. K. Cui [13], leaf spots and new shoot canker caused by *Fusicoccum cryptomeriae* Sawada [14], and the shoot blight caused by *Pestalotiopsis neglecta* Thüm [15]. The occurrence of the disease has detrimentally affected the ecological functions and economic value of *C. japonica* and restricted the development of this species.

*Neofusicoccum* was proposed by Crous et al. in 2006 with the type species of *Neofusicoccum parvum* (Pennycook & Samuels) Crous, Slippers & A.J.L. Phillips [16]. It belongs to Botryosphaericeae, and the fungi of this family include various non-host-specific pathogens, saprobes, endophytes, and potential pathogens [17]. In the past several years, diseases caused by *Neofusicoccum* species were frequently reported in many countries. *N. parvum* was reported as a pathogen of many plants, including the stem and branch blight disease of *Zanthoxylum bungeanum* in Sichuan, China [18], nut rot of chestnut (*Castanea sativa*) in Italy [19], and dieback and canker of hemp (*Cannabis sativa* L.) in the United States [20]. *N. luteum* (Pennycook & Samuels) Crous & al., *N. batangarum* Begoude & al., *N. mangiferae* Syd. & P. Syd, and some other species in *Neofusicoccum* were also reported to relate to some plant diseases [21,22,23]. Due to the similarity of morphological characteristics in *Neofusicoccum*, some species were erroneously classified in the past. With the development of phylogenetic analysis, many new species were found, and some known species were reclassified. Five novel species (*N. dianense* G.Q. Li & S.F. Chen, *N. magniconidium* G.Q. Li & S.F. Chen, *N. ningerense* G.Q. Li & S.F. Chen, *N. parviconidium* G.Q. Li & S.F. Chen) were described by Li et al. [24]. In a recent taxonomic study of Botryosphaericeae, 11 species in *Neofusicoccum* were reduced to synonymy, and 2 novel species, named *N. podocarpi* W. Zhang & Crous and *N. rapaneae* W. Zhang & Crous, were described [25]. Since then, several studies have been published and described a number of new taxa, such as *Neofusicoccum caryigenum* M.T. Brewer & C.J. Cameron [26], *Neofusicoccum sichuanense* X. L. Xu & C. L. Yang [27], *N. hyperici* Y. Hattori & C. Nakash, *N. miyakoense* Y. Hattori & C. Nakash, and *N. okinawaense* Y. Hattori & C. Nakash [28], *Neofusicoccum moracearum* Tennakoon, C.H. Kuo & K.D. Hyde [29], and *Neofusicoccum mystacidii* Crous [30]. At present, there are 52 species of *Neofusicoccum* [25,27,28].

Recently, during disease surveys in Nanjing, Jiangsu Province, China, a new disease of *C. japonica* was found. A total of 130 trees were surveyed and more than 90% showed the same symptom of dieback of the stem and branch. It is different from the other reported diseases of *C. japonica*. The infected plants showed an obvious distinction in the conjunct area of healthy and infected parts, both in color and texture in the xylem and the interior side of the bark. The occurrence of the disease seriously damaged the ornamental value of *C. japonica*. The aims of this study were to: (1) test the pathogenicity of representative isolates, and (2) determine the identities of the fungi causing stem basal canker of *C. japonica* based on morphological characteristics and phylogenetic analyses and describe a new fungal species, which is also pathogenic to *C. japonica*.

## 2. Materials and Methods

### 2.1. Sample Collection and Fungal Isolation

Disease surveys were conducted at Nanjing Forestry University from September to October 2020. Approximately 50 bark pieces from the edges of both healthy and infected trunks were collected from 3 symptomatic trees. The interior side of the bark was cut into small pieces (3–5 mm^2^) using a sterile scalpel. The pieces were submerged in 75% ethanol for 30 s and then in 1.5% NaClO for 90 s, washed 3 times in sterile water, blotted dry with sterilized filter paper, and placed onto potato dextrose agar (PDA) with 100 µg/mL of ampicillin (Nanjing Zebra Experimental Equipment Co., Ltd., Nanjing, China). Cultures were incubated at 25 °C in the dark for five days, and hyphae tips at the edge of the colony were transferred to the new PDA plates.

### 2.2. Pathogenicity Tests

To determine the pathogenicity of the isolates on *Cryptomeria japonica*, the trunk of two-year-old seedlings was wounded with a sterile scalpel to expose the cambium. The wounds were located approximately 2 cm above the soil level. For inoculation, 5 mm plugs were cut out from the growing edges of 5-day-old cultures placed into the wounds with the mycelia facing the cambium, and then the inoculating site was wrapped with Parafilm. Thirteen representative fungal isolates (G1, G2, G4, G11, G15, G16, G18, G23, G24, G74, G91, G92, and C7) were used for inoculation, and 3 to 5 seedlings were inoculated with each isolate. Five control seedlings were treated with sterile PDA agar using the same aforementioned method [31]. All inoculated plants were kept in a greenhouse (relative humidity > 80%, 25 ± 2 °C). The inoculated fungi were re-isolated as described above and confirmed by both morphological characteristics and ITS sequence analysis.

### 2.3. Morphological Identification

After the pathogenicity experiment, seven pathogenic isolates were cultured on PDA for seven days, and the colony color, texture, and pigment production of the isolates were observed and recorded. To induce sporulation, the fungi were cultured on Petri dishes containing synthetic nutrient-poor agar medium (SNA) or 2% tap water agar (WA) supplemented with double autoclaved pine needles on their surface [32,33,34]. All colonies were placed under near-ultraviolet light at 25 °C for 1–2 months. The morphology and size of 30 pycnidia of each isolate were observed and recorded using a Zeiss stereomicroscope (SteRo Discovery v20). Relevant morphological characteristics were observed and recorded using a Zeiss Axio Imager A2m microscope. The lengths, widths, and shapes of 20 conidiophores, 20 conidiogenous cells, and 50 conidia of the 7 isolates were measured [32].

### 2.4. DNA Extraction, PCR Amplification, and Sequencing

Total genomic DNA from isolates in this study was extracted from the fungal mycelia of 7-day-old cultures using the cetyltrimethylammonium bromide (CTAB) protocol. Four loci, including internal transcribed spacer (ITS), partial translation elongation factor 1-alpha (*tef1*), partial β-tubulin (*tub2*), and partial DNA-directed RNA polymerase II subunit (*rpb2*), were amplified with the primer pairs ITS1/ITS4 [35], EF1-728F/EF1-986R [36], BT-2a/BT-2b [37], and RPB2bot6F/RPB-2bot7R [38,39], respectively. The polymerase chain reaction consisted of 25 μL of Taq DNA polymerase mix, 2 μL of genomic DNA, 2 μL of each primer, and 19 μL of double-distilled water. The amplification conditions consisted of an initial denaturation step at 95 °C for 5 min, 34 cycles of 95 °C for 30 s, and annealing at a suitable temperature for 30 s for each locus: 56 °C (ITS), 52 °C (*tef1*), 60 °C (*tub2*), and 55 °C (*rpb2*), and then 72 °C for 30 s, followed by a final elongation step at 72 °C for 10 min and a cool-down step to 4 °C. Primers were synthesized and PCR products were sequenced by the Shanghai Jieli Biotechnology Co. Ltd., Nanjing, Jiangsu Province, China. All sequences of the isolates from this study were deposited in GenBank (http://www.ncbi.nlm.nih.gov (accessed on 28 July 2022)) (Table 1).

### 2.5. Phylogenetic Analyses

Initial identities of the isolates were determined using BLASTn of the NCBI GenBank with sequences generated in this study. ITS, *tef1*, *tub2*, and *rpb2* sequences of phylogenetically related *Neofusicoccum* species and *Botryosphaeria dothidea* (CBS 115476) as an outgroup were obtained from GenBank (Table 1). The sequences of *Neofusicoccum* isolates obtained during this study were aligned based on loci with reference sequences, respectively, in PhyloSuite V1.2.2 using the ‘FFT-NS-2 (default)’ strategy and normal alignment mode of MAFFT V7.313, and then edited manually where necessary [52,53]. Two phylogenetic analyses were conducted using IQtree ver. 1.6.8 for the maximum likelihood (ML) analysis and MrBayes 3.2.6 for Bayesian Inference (BI) analysis [54,55]. ModelFinder was used to select the best-fit model for the multi-locus phylogenetic analyses [56]. For ML analysis, with 1000 bootstrap replicates, we utilized the best-fit model: GTR + F + I + G4. For BI analysis, we used the GTR + I + G + F model (2 parallel runs, 2,000,000 generations), in which the initial 25% of sampled data were discarded as burn-in. All phylogenetic trees were viewed using FigTree v. 1.4.4 (http://tree.bio.ed.ac.uk/software/figtree/ (accessed on 12 February 2023)).

### 2.6. Genealogical Concordance Phylogenetic Species Recognition Analysis

The concatenated dataset (ITS, *tef1*, *tub2*, and *rpb2*) was used to analyze the new species, their species limits, and their most closely related taxa, as described by Quaedvlieg et al. [57], through the GCPSR concept with a PHI test performed in SplitsTree v.4.14.6. A PHI index below 0.05 (Φ_w_ < 0.05) indicates the presence of significant recombination in the dataset. The relationships between this new taxon and closely related species were visualized in splits graphs with both the LogDet transformation and splits decomposition options.

## 3. Results

### 3.1. Symptoms in the Field and Fungal Isolation

The results of drone aerial photography and field investigation showed that 90 percent of *C. japonica* showed symptoms of dieback (130 trees in total) (Figure 1A,B). Affected trees have normal bark and reddish-brown canopies (Figure 1C,D). After stripping the bark, lesions can be observed in the phloem, which spread from the root color to the breast height of the trunk (Figure 1E–H). The lesions were dry and brownish, and the healthy xylem was yellowish and moist. A total of 157 fungal isolates were isolated and were divided into 6 groups according to the colony morphology. Thirteen representative isolates (G1, G2, G4, G11, G15, G16, G18, G23, G24, G74, G91, G92, and C7) were selected for pathogenicity experiments.

### 3.2. Pathogenicity Assays

Three days after the inoculation, brown spots began to appear at the inoculation points of isolates G1, G2, G15, G16, G24, G91, and G92. The lesions gradually expanded in about a week, and after 20 days, half of the whole plant withered from the bottom to the top, and the whole plant withered in 1 month (Figure 2B–H). These symptoms were consistent with those observed in the field. At the same time, the control, G4, G11, G18, G23, G74, and C7 did not develop symptoms (Figure 2A). The inoculated fungal isolates were re-isolated from the lesions on the inoculated seedlings, and no fungi were isolated from the control. Therefore, the seven isolates (G1, G2, G15, G16, G24, G91, and G92) were identified as the causal agents of stem basal canker on *C. japonica*.

### 3.3. Molecular Identification of the Fungal Isolates

Phylogenetic analyses showed that four isolates (G15, G16, G91, and G92) were in the same cluster with *N. parvum* (ex-type: ATCC 58191). Three isolates (G1, G2, and G24) were clustered in a distinct clade, which was distinct from all other known species and a sister clade to the clade of *N. sinense* (ex-type: CGMCC 3.18315) (Figure 3). Based on the phylogenetic analyses using the concatenated sequences of the ITS, *tef1*, *tub2*, and *rpb2* sequences, four isolates (G15, G16, G91, and G92) were *N. parvum*, and three isolates (G1, G2, and G24) were a new species of *Neofusicoccum*. The tree topologies of ML and BI phylogenetic trees were consistent, where maximum likelihood bootstrap support values (ML ≥ 50) and Bayesian posterior probability (PP ≥ 0.90) are shown at the nodes (ML/PP). Furthermore, the PHI test on *N. cryptomeriae* revealed that there was no significant recombination (Φ_w_ = 0.163) among their closely related taxa: *N. sinense*, *N. brasiliense*, and *N. kwambonambiense* (Figure 4). Thus, the isolates G1, G2, and G24 were confirmed to be new species.

### 3.4. Morphology and Taxonomy

For isolates G1, G2, and G24, morphological differences were observed compared to the most closely related species (*Neofusicoccum sinense* CGMCC 3.18315) based on phylogenetic analyses. Therefore, the results of the phylogenetic analyses and morphological studies support the conclusion that three isolates (G1, G2, and G24) were a *Neofusicoccum* species new to science. The new species is described as follows:

*Neofusicoccum cryptomeriae* Li-Hua Zhu, Yuan-Zhi Si, Jian-Wei Sun & D. W. Li, sp. nov. (Figure 5).

Index Fungorum number, IF 900283.

Etymology. Latin: *cryptomeriae* referring to the host genus *Cryptomeria*.

Sexual state: Undetermined. Asexual state: Conidiomata pycnidial, produced on pine needles on SNA within 30 days, solitary or aggregated, covered by mycelium, dark-brown to black, stylolitic, ellipsoidal or spherical, up to 183–381 μm-wide, and 463–1152 μm-high. Conidiophores hyaline, cylindrical, branched, and smooth: (20.6–)25–34.1(–39) × (3.1–)3.6–3.8(–5.2) μm (av. = 29.6 × 4.2 μm, n = 20). Conidiogenous cells holoblastic, hyaline, cylindrical, and phialidic, with periclinal thickening: (11.6–)13.8–22.4(–25.8) × (3–)3.4–4.2(–4.9) μm (av. = 18.1 × 3.8 μm, n = 20). Paraphyses not observed. Conidia 1-celled, hyaline, thin-walled, smooth with granular contents, fusiform, initially non-septate, and subsequently becoming 1–2 septate: (20.9–)23.0–26.1(–27.5) × (6.8–)7.0–7.8(–8.3) μm (av. = 24.6 × 7.4 μm, n = 50; L/W = 3.3).

Culture characteristics: Colonies on PDA were initially white with fluffy mycelia. After five days, the aerial mycelia were dense at the edge of the colony and sparse in the middle, and hyphae at the center of the front and back sides are gray.

The measured data of isolates G1 and G2 are listed in Table 2.

Holotype: China, Jiangsu, Nanjing, 32°04′53.11″ N, 118°49′10.27″ E, isolated from *Cryptomeria japonica*, 2 September 2020, Jian-Wei Sun, CFCC 55721 (=G24). The holotype specimen is a living specimen being maintained via lyophilization at the China Forestry Culture Collection Center (CFCC), Chinese Academy of Forestry, Beijing, China.

Additional materials examined: China, Jiangsu, Nanjing, 32°04′53.11″ N, 118°49′10.27″ E, isolated from *Cryptomeria japonica*, 2 September 2020, Jian-Wei Sun, CFCC 55720 (=G1). China, Jiangsu, Nanjing, 32°04′53.11″ N, 118°49′10.27″ E, isolated from *Cryptomeria japonica*, 2 September 2020, Jian-Wei Sun, CFCC 55728 (=G2).

Host/distribution: from *Cryptomeria japonica* in Nanjing, Jiangsu, China.

Notes: Phylogenetically, *N. cryptomeriae* is closely related to *N. sinense*. They were distinguished based on 14 nucleotides in the concatenated alignment, of which 9 were distinct in ITS, 4 in *tef1*, and 1 in *tub2*. *N. sinense* has no sequence data of *rpb2* for comparison. Morphologically, *N. cryptomeriae* differs from *N. sinense* by its longer conidia (23–26.1 × 7–7.8 μm vs. 17.6–20.4 × 7.4–8 μm) (Table 2). *Fusicoccum cryptomeriae* was a pathogen causing leaf spots on *C. japonica*, and it was differentiated from *N. cryptomeriae* by its much smaller conidia: 6.5–8 × 2.5 μm vs. 23.0–26.1 × 7.0–7.8 μm [14].

*Neofusicoccum parvum* (Pennycook & Samuels) Crous, Slippers & A.J.L. Phillips.

Based on analyses of DNA sequence data, four isolates (G15, G16, G91, and G92) were in the same cluster with *N. parvum*. Comparing these four isolates with the morphological characteristics of *N. parvum*, including colony, conidiomata, conidiophores, conidiogenous cells, and conidia, showed that the morphologies of the four isolates were the same as *N. parvum*. Therefore, these four isolates were *N. parvum*. The morphological characteristics of the representative isolate G15 are as follows (Figure 6). On PDA, G15 was initially white, and after 5 days, it developed an abundant greyish-white aerial mycelium. Conidiomata pycnidial, produced on pine needles on WA within 30 days, solitary or in groups, covered by mycelium, dark-brown to black, up to 172–247 μm-wide, and 144–440-μm high. Conidiogenous cells were hyaline and short subcylindrical: (11.4–)13.4–18.4(–19.4) × (2.5–)2.6–3.8(–4.2) μm (av. = 15.9 × 3.2 μm, n = 20). Conidia 1-celled, hyaline, ellipsoidal to fusiform: (15.5–)17–18.7(–20.3) × (5.2–)5.8–6.3(–6.8) μm (av. = 18.3 × 6.2 μm, n = 50; L/W = 3), and when mature, became brown, septate. The measured data of isolates G16, G91, and G92 are listed in Table 2.

## 4. Discussion

In this study, the pathogen causing stem basal canker of *C. japonica* in Nanjing, China, was determined by the pathogenicity tests using Japanese cedar seedlings. Based on morphological, GCPSR principle, and phylogenetic studies, the pathogens were identified as two species of *Neofusicoccum*, *N. parvum* and a new species, *Neofusicoccum cryptomeriae*.

As early as 2013, studies have shown that *N. parvum* is a widely distributed and common pathogen to plants, occurring on 90 host species across 6 continents [58]. At present, this species has been reported in 181 plants [59]. In subsequent studies, *N. parvum* was found to be one of the most virulent species based on the extent of necroses it causes [60,61,62,63]. *Neofusicoccum* spp. have not been reported to infect Japanese cedar, but many species of *Neofusicoccum* have been reported on other conifers, including many endangered species. For example, *N. nonquaesitum* was reported to cause branch cankers on *Sequoiadendron giganteum* in North America [64]. *N. parvum* was reported to cause canker and dieback of *S. giganteum* in the Geneva Lake area, Switzerland [65]. *N. nonquaesitum* has also been reported to cause branch dieback and decline in *Araucaria araucana*, and this tree species has been listed as an endangered species on the Red Data List of the International Union for Conservation of Nature [66].

The asexual state developed under natural conditions is very important for the morphological identification of fungi [67]. Many genera of Botryosphaericeae, including *Neofusicoccum*, *Botryosphaeria*, *Pseudofusicoccum*, and *Neoscytalidium,* share similar morphological characteristics of their asexual states, and most of their conidia are narrow ellipsoids [16]. Similar morphological characteristics make it difficult to differentiate the genera. Botryosphaericeae can grow well on culture media, but it is difficult to produce conidia [16]. These conditions have led to some challenges in the morphological identification of fungi in this family. Many species of *Neofusicoccum* are similar in morphology, and the molecular methods provide supplementary methods of fungal identification and classification. In recent years, phylogenetic and phytopathological studies on *Neofusicoccum* have used concatenated sequences of ITS, *tef1*, *tub2*, and *rpb2* [24,28,68]. Phylogenetic analyses using sequences of multiple loci can better distinguish and identify some closely related species in the genus.

The research on *Neofusicoccum* fungi was mainly focused on the identification of plant pathogens in China, where there is a lack of systematic taxonomic research. New species of this genus have been continuously discovered, indicating that the fungal resources of the genus *Neofusicoccum* are relatively abundant in China. It is necessary to collect a large number of specimens to establish a complete and reasonable classification system for this genus to provide mycological and molecular information and a scientific basis for disease prevention and control.

## Figures and Tables

**Figure 1 jof-09-00404-f001:**
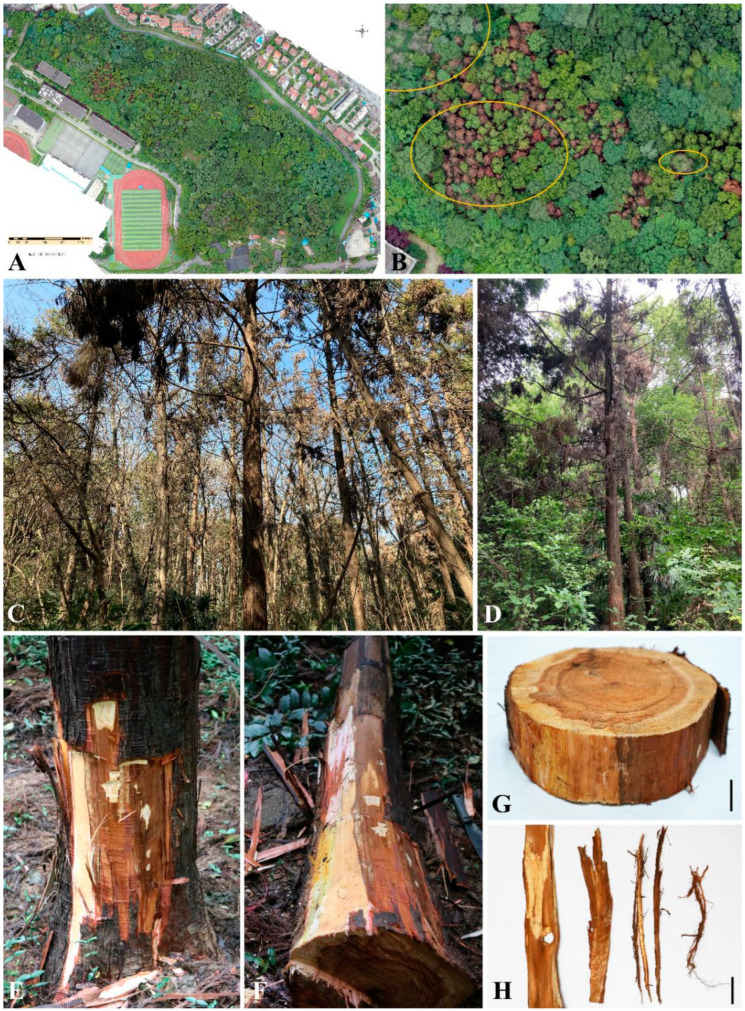
Symptoms of stem basal canker disease on *C. japonica* in the field. (**A**,**B**) Aerial view. (**C**,**D**) Stem basal canker of *C. japonica*. (**E**–**G**) Lesions on the phloem of the tree trunk. (**H**) Lesions on the phloem of the tree root. Scale bars: (**G**,**H**) = 5 cm.

**Figure 2 jof-09-00404-f002:**
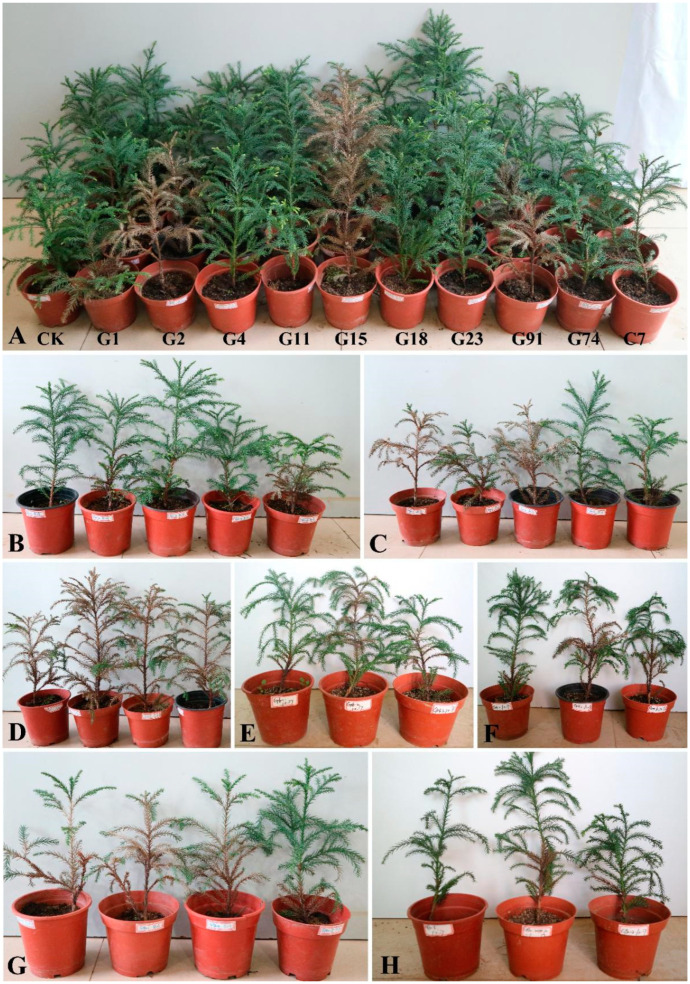
Symptoms caused by fungal isolates in this study 20 days after inoculation. (**A**) Control, G4, G11, G18, G23, G74, and C7, showing the absence of lesion development on *C. japonica*. G1, G2, G15, and G92, showing the lesion development on *C. japonica*. (**B**–**H**) Lesions produced on *C. japonica* by isolates (**B**) G1, (**C**) G2, (**D**) G15, (**E**) G16, (**F**) G24, (**G**) G91, and (**H**) G92.

**Figure 3 jof-09-00404-f003:**
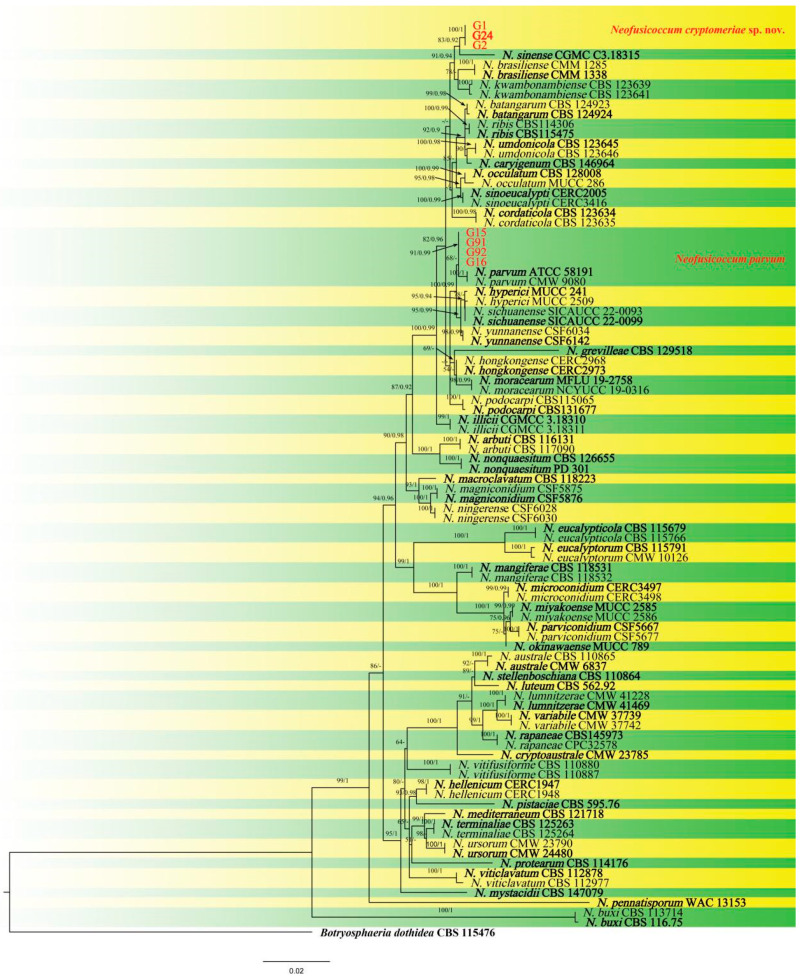
Phylogenetic relationship of *Neofusicoccum cryptomeriae* and *N. parvum* with closely related taxa derived from a maximum likelihood (ML) analysis and Bayesian Inference using combined ITS, *tef1*, *tub2,* and *rpb2* sequence alignment, with *Botryosphaeria dothidea* (CBS 115476) as the outgroup. Maximum likelihood bootstrap support values (ML ≥ 50) and Bayesian posterior probability (PP ≥ 0.90) are shown at the nodes (ML/PP). Ex-type strains are marked in bold, the species are delimited with colored blocks, and isolates in this study are marked in red.

**Figure 4 jof-09-00404-f004:**
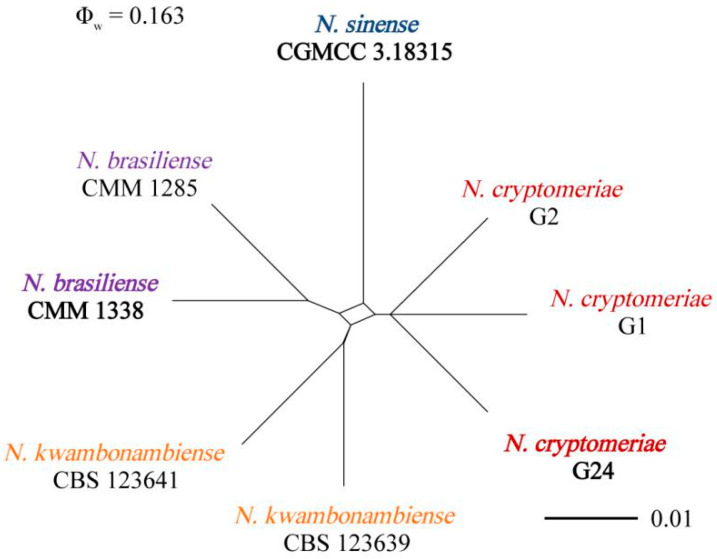
Pairwise homoplasy index (PHI) test of *Neofusicoccum cryptomeriae* and closely related *N. sinense*, *N. brasiliense*, and *N. kwambonambiense* using both LogDet transformation and splits decomposition. PHI test results (Φ_w_) < 0.05 indicate significant recombination within the dataset.

**Figure 5 jof-09-00404-f005:**
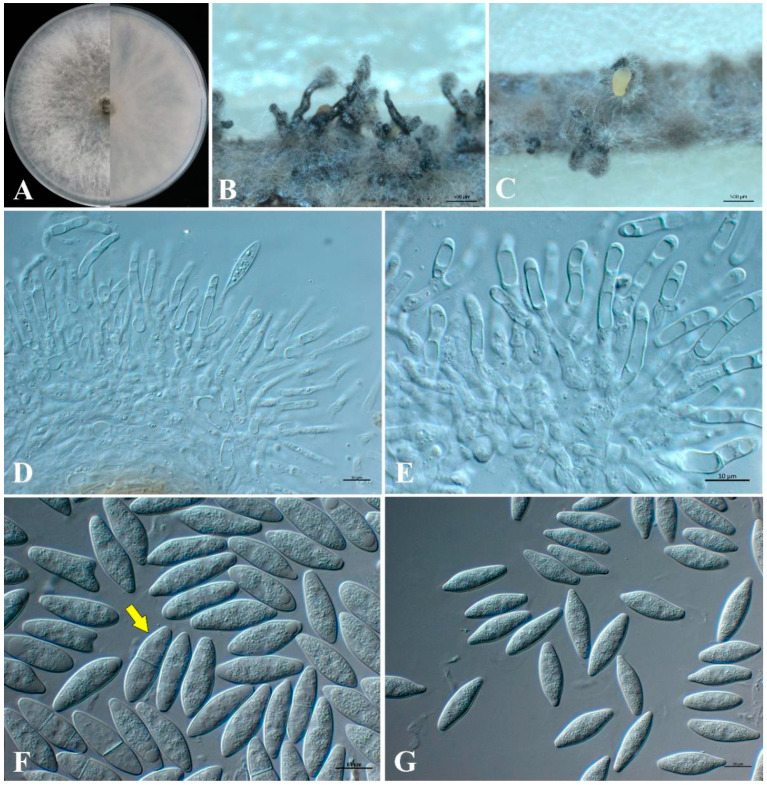
Morphology of *Neofusicoccum cryptomeriae* (G24). (**A**) Five-day-old front and back view culture on PDA. (**B**,**C**) Conidioma formed on pine needle culture. (**D**) Conidiophores, conidiogenous cells, and developing conidia. (**E**) Conidiogenous cells. (**F**) Conidia with 1 septum (indicated by arrow). (**G**) Conidia. Scale bars: (**B**,**C**) = 500 μm; (**D**–**G**) = 10 μm.

**Figure 6 jof-09-00404-f006:**
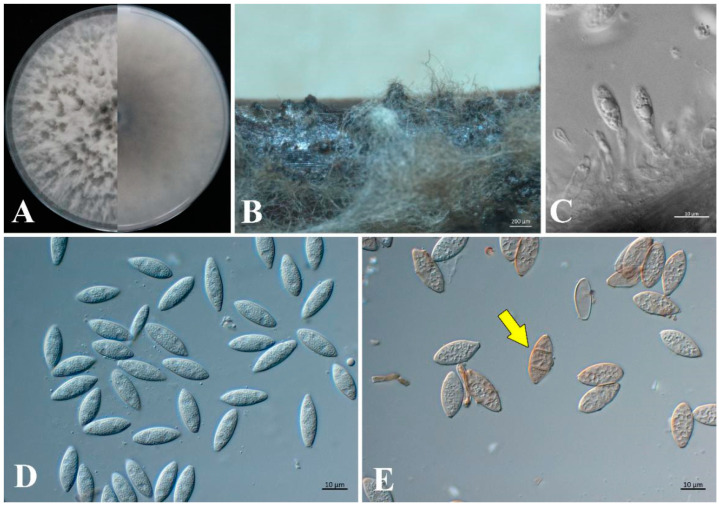
Morphology of *Neofusicoccum parvum* (G15). (**A**) Five-day-old front and back view culture on PDA. (**B**) Conidioma formed on pine needle culture. (**C**) Conidiogenous cells and developing conidia. (**D**) Hyaline conidia. (**E**) Brown, septate conidia (indicated by arrow). Scale bars: (**B**) = 200 μm; (**C**–**E**) = 10 μm.

**Table 1 jof-09-00404-t001:** Strains obtained in this study and downloaded from GenBank with accession numbers used for phylogenetic analyses.

		GenBank Accession Number				
Species	Isolate	ITS ^c^	*tef1* ^c^	*tub2* ^c^	*rpb2* ^c^	Reference
** *Neofusicoccum arbuti* **	**CBS 116131 ^a^**	**AY819720**	**KF531792**	**KF531793**	**KX464003**	[32]
*N. arbuti*	CBS 117090	AY819724	KF531791	KF531794	N/A	[32]
*N. australe*	CBS 110865	AY343408	KX464661	KX464937	KX464005	[40]
** *N. australe* **	**CMW 6837 ^a^**	AY339262	AY339270	AY339254	EU339573	[40]
*N. batangarum*	CBS 124923	FJ900608	FJ900654	FJ900635	FJ900616	[41]
** *N. batangarum* **	**CBS 124924 ^a^**	**FJ900607**	**FJ900653**	**FJ900634**	**FJ900615**	[41]
*N. brasiliense*	CMM 1285	JX513628	JX513608	KC794030	N/A	[42]
** *N. brasiliense* **	**CMM 1338 ^a^**	**JX513630**	**JX513610**	**KC794031**	**N/A**	[42]
*N. buxi*	CBS 113714	KX464164	KX464677	KX464954	KX464009	[40]
** *N. buxi* **	**CBS 116.75 ^a^**	**KX464165**	**KX464678**	**N/A**	**KX464010**	[40]
** *N.caryigenum* **	**CBS 146964 ^a^**	**MW405114**	**MW393657**	**MW393679**	**MW393668**	[26]
** *N. cordaticola* **	**CBS 123634 ^a^**	**EU821898**	**EU821868**	**EU821838**	**EU821928**	[38]
*N. cordaticola*	CBS 123635	EU821903	EU821873	EU821843	EU821933	[38]
** *N. cryptoaustrale* **	**CMW 23785 ^a^**	**FJ752742**	**FJ752713**	**FJ752756**	**KX464014**	[40]
** *N. cryptomeriae* **	**G24 = CFCC 55721 ^a,b^**	**ON209700**	**OP056461**	**OP056458**	**OP056455**	this study
*N. cryptomeriae*	G1 = CFCC 55720 ^b^	**ON209698**	**OP056459**	**OP056456**	**OP056453**	this study
*N. cryptomeriae*	G2 = CFCC 55728 ^b^	**ON209699**	**OP056460**	**OP056457**	**OP056454**	this study
** *N. eucalypticola* **	**CBS 115679 ^a^**	**AY615141**	**AY615133**	**AY615125**	**N/A**	[43]
*N. eucalypticola*	CBS 115766	AY615143	AY615135	AY615127	N/A	[43]
** *N. eucalyptorum* **	**CBS 115791 ^a^**	**AF283686**	**AY236891**	**AY236920**	**N/A**	[44]
*N. eucalyptorum*	CMW 10126	AF283687	AY236892	AY236921	N/A	[44]
** *N. grevilleae* **	**CBS 129518 ^a^**	**JF951137**	**N/A**	**N/A**	**N/A**	[45]
** *N. hellenicum* **	**CERC1947 ^a^**	**KP217053**	**KP217061**	**KP217069**	**N/A**	[46]
*N. hellenicum*	CERC1948	KP217054	KP217062	KP217070	N/A	[46]
*N. hongkongense*	CERC2968	KX278051	KX278156	KX278260	KX278282	[17]
** *N. hongkongense* **	**CERC2973 ^a^**	**KX278052**	**KX278157**	**KX278261**	**KX278283**	[17]
** *N. hyperici* **	**MUCC 241 ^a^**	**LC589125**	**LC589137**	**LC589147**	**LC589160**	[28]
*N. hyperici*	MUCC 2509	LC589126	LC589138	LC589148	LC589161	[28]
** *N. illicii* **	**CGMCC 3.18310 ^a^**	**KY350149**	**N/A**	**KY350155**	**N/A**	[34]
*N. illicii*	CGMCC 3.18311	KY350150	KY817756	KY350156	N/A	[34]
*N. kwambonambiense*	CBS 123639	EU821900	EU821870	EU821840	EU821930	[38]
*N. kwambonambiense*	CBS 123641	EU821919	EU821889	EU821859	EU821949	[38]
*N. lumnitzerae*	CMW 41228	KP860882	KP860725	KP860803	KU587926	[47]
** *N. lumnitzerae* **	**CMW 41469 ^a^**	**KP860881**	**KP860724**	**KP860801**	**KU587925**	[47]
** *N. luteum* **	**CBS 562.92 ^a^**	**KX464170**	**KX464690**	**KX464968**	**KX464020**	[40]
** *N. macroclavatum* **	**CBS 118223 ^a^**	**DQ093196**	**DQ093217**	**DQ093206**	**KX464022**	[48]
*N. magniconidium*	CSF5875	MT028611	MT028777	MT028943	MT029084	[24]
** *N. magniconidium* **	**CSF5876 ^a^**	**MT028612**	**MT028778**	**MT028944**	**MT029085**	[24]
** *N. mangiferae* **	**CBS 118531 ^a^**	**AY615185**	**DQ093221**	**AY615172**	**N/A**	[48]
*N. mangiferae*	CBS 118532	AY615186	DQ093220	AY615173	KX464023	[48]
** *N. mediterraneum* **	**CBS 121718 ^a^**	**GU251176**	**GU251308**	**GU251836**	**KX464024**	[49]
** *N. microconidium* **	**CERC3497 ^a^**	**KX278053**	**KX278158**	**KX278262**	**MF410203**	[17]
*N. microconidium*	CERC3498	KX278054	KX278159	KX278263	MF410204	[17]
** *N. miyakoense* **	**MUCC 2585 ^a^**	**N/A**	**LC589146**	**LC589157**	**LC589170**	[28]
*N. miyakoense*	MUCC 2586	LC589133	LC589144	LC589155	LC589168	[28]
** *N. moracearium* **	**MELU 19-2758 ^a^**	**NR174834**	**MW183808**	**N/A**	**N/A**	[29]
*N. moracearium*	MFLU 19-0316	MW063187	MW183809	N/A	N/A	[29]
** *N. mystacidii* **	**CBS 147079 ^a^**	**NR173012**	**MW890094**	**MW890133**	**MW890065**	[30]
*N. ningerense*	CSF6028	MT028613	MT028779	MT028945	MT029086	[24]
*N. ningerense*	CSF6030	MT028614	MT028780	MT028946	MT029087	[24]
** *N. nonquaesitum* **	**CBS 126655 ^a^**	**GU251163**	**GU251295**	**GU251823**	**KX464025**	[49]
*N. nonquaesitum*	PD 301	GU251164	GU251296	GU251824	N/A	[49]
** *N. occulatum* **	**CBS 128008 ^a^**	**EU301030**	**EU339509**	**EU339472**	**EU339558**	[39]
*N. occulatum*	MUCC 286	EU736947	EU339511	EU339474	EU339560	[39]
** *N. okinawaense* **	**MUCC 789 ^a^**	**LC589134**	**LC589145**	**LC589156**	**LC589169**	[28]
** *N. parviconidium* **	**CSF5667 ^a^**	**MT028615**	**MT028781**	**MT028947**	**MT029088**	[24]
*N. parviconidium*	CSF5677	MT028619	MT028785	MT028951	MT029092	[24]
** *N. parvum* **	**ATCC 58191 ^a^**	**AY236943**	**AY236888**	**AY236917**	**EU821963**	[44]
*N. parvum*	CMW 9080	AY236942	AY236887	AY236916	EU821962	[44]
*N. parvum*	G15 = CFCC 55724 ^b^	ON209685	OP095379	OP095383	OP095387	this study
*N. parvum*	G16 = CFCC 55718 ^b^	ON209686	OP095380	OP095384	OP095388	this study
*N. parvum*	G91 = CFCC 55719 ^b^	ON209687	OP095381	OP095385	OP095389	this study
*N. parvum*	G92 = CFCC 55723 ^b^	ON209688	OP095382	OP095386	OP095390	this study
** *N. pennatisporum* **	**WAC 13153 ^a^**	**NR136987**	**EF591976**	**EF591959**	**N/A**	[50]
** *N. pistaciae* **	**CBS 595.76 ^a^**	**KX464163**	**KX464676**	**KX464953**	**KX464008**	[40]
*N. podocarpi*	CBS 115065	MT587507	MT592222	MT592714	MT592411	[25]
** *N. podocarpi* **	**CBS 131677 ^a^**	**MT587508**	**MT592223**	**MT592715**	**MT592412**	[25]
** *N. protearum* **	**CBS 114176 ^a^**	**AF452539**	**KX464720**	**KX465006**	**KX464029**	[40]
** *N. rapaneae* **	**CBS 145973 ^a^**	**MT587511**	**MT592226**	**MT592718**	**MT592415**	[25]
*N. rapaneae*	CPC 32578	MT587512	MT592227	MT592719	MT592416	[25]
*N. ribis*	CBS 114306	MT587514	MT592229	MT592721	MT592418	[25]
** *N. ribis* **	**CBS 115475 ^a^**	**AY236935**	**AY236877**	**AY236906**	**EU821958**	[25]
** *N. sichuanense* **	**SICAUCC** **22-0099 ^a^**	**OP058990**	**OP066336**	**OP066363**	**OP066355**	[27]
*N. sichuanense*	SICAUCC22-0093	OP058984	OP066333	OP066357	OP066349	[27]
** *N. sinense* **	**CGMCC 3.18315 ^a^**	**KY350148**	**KY817755**	**KY350154**	**N/A**	[34]
** *N. sinoeucalypti* **	**CERC2005 ^a^**	**KX278061**	**KX278166**	**KX278270**	**KX278290**	[17]
*N. sinoeucalypti*	CERC3416	KX278064	KX278169	KX278273	KX278293	[17]
** *N. stellenboschiana* **	**CBS 110864 ^a^**	**AY343407**	**AY343348**	**KX465047**	**KX464042**	[40]
** *N. terminaliae* **	**CBS 125263 ^a^**	**GQ471802**	**GQ471780**	**KX465052**	**KX464045**	[40]
*N. terminaliae*	CBS 125264	GQ471804	GQ471782	KX465053	KX464046	[40]
** *N. umdonicola* **	**CBS 123645 ^a^**	**EU821904**	**EU821874**	**EU821844**	**EU821934**	[40]
*N. umdonicola*	CBS 123646	EU821905	EU821875	EU821845	EU821935	[40]
*N. ursorum*	CMW 23790	FJ752745	FJ752708	KX465057	N/A	[40]
** *N. ursorum* **	**CMW 24480 ^a^**	**FJ752746**	**FJ752709**	**KX465056**	**KX464047**	[40]
** *N. variabile* **	**CMW 37739 ^a^**	**MH558608**	**N/A**	**MH569153**	**N/A**	[51]
*N. variabile*	CMW 37742	MH558609	MH576585	MH569154	N/A	[51]
** *N. viticlavatum* **	**CBS 112878 ^a^**	**AY343381**	**AY343342**	**KX465058**	**KX464048**	[32]
*N. viticlavatum*	CBS 112977	AY343380	AY343341	KX465059	N/A	[32]
*N. vitifusiforme*	CBS 110880	AY343382	AY343344	KX465008	N/A	[40]
*N. vitifusiforme*	CBS 110887	AY343383	AY343343	KX465061	KX464049	[40]
*N. yunnanense*	CSF6034	MT028672	MT028838	MT029004	MT029117	[24]
** *N. yunnanense* **	**CSF6142 ^a^**	**MT028667**	**MT028833**	**MT028999**	**MT029112**	[24]
** *Botryosphaeria dothidea* **	**CBS 115476 ^a^**	**AY236949**	**AY236898**	**AY236927**	**EU339577**	[44]

^a^ Ex-type cultures are shown in bold. ^b^ Isolates used in this study. ^c^ ITS, internal transcribed spacer; *rpb2*, DNA-directed RNA polymerase II subunit; *tef1*, partial translation elongation factor 1-alpha gene; *tub2*, partial beta-tubulin gene.

**Table 2 jof-09-00404-t002:** Synoptic characters of two species of *Neofusicoccum* in this study and related *Neofusicoccum* spp.

Species	Isolates	Conidiogenous Cells (μm)	Conidia (μm)	Reference
*Neofusicoccum cryptomeriae*	G1 = CFCC 55720	16.4–24.1 × 3–3.9	20.5–25.7 × 7.1–8.9	this study
	G2 = CFCC 55728	15.8–24.2 × 3.2–4	20.1–22.6 × 6.8–8	this study
	G24 = CFCC 55721	13.8–22.4 × 3.4–4.2	23–26.1 × 7–7.8	this study
*N. parvum*	G15 = CFCC 55724	13.4–18.4 ×2.6–3.8	17–18.7 × 5.8–6.3	this study
	G16 = CFCC 55718	11.4–19.5 × 2.1–3.6	16.8–18.1 × 5.7–6.3	this study
	G91 = CFCC 55719	13.7–20.9 × 2.5–3.3	17.1–19.5 × 5.7–6.4	this study
	G92 = CFCC 55723	12.2–21.4 × 2.6–3.4	17.1–18.8 × 5.6–6.3	this study
*N. sinense*	CGMCC 3.18315	not observed	17.6–20.4 × 7.4–8	[34]

## Data Availability

All sequences generated in this study were submitted to GenBank.

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
