# Peer review of "Neofusicoccum cryptomeriae sp. nov. and N. parvum Cause Stem Basal Canker of Cryptomeria japonica in China"

_jof, 2023, doi:10.3390/jof9040404_

Round 1
Reviewer 1 Report
This is an interesting paper.I think authors have worked on it rather well.
They have characterised the fungal pathogens based on a polyphasic approach.The writing (English) is also good, something i seldom see with chinese authors (especially those who are not well versed in writing english scientific papers) but this one is good.
The analyses done are up to standard and the experiment has been well designed. When it comes to establishment of the new species, perhaps it would be good to comment on the differences in DNA sequences across the different gene regions (% differences, similarities).
Author Response
Response to Reviewer 1 Comments
Point 1: This is an interesting paper.I think authors have worked on it rather well.
They have characterised the fungal pathogens based on a polyphasic approach.The writing (English) is also good, something i seldom see with chinese authors (especially those who are not well versed in writing english scientific papers) but this one is good.
The analyses done are up to standard and the experiment has been well designed. When it comes to establishment of the new species, perhaps it would be good to comment on the differences in DNA sequences across the different gene regions (% differences, similarities).
Response 1: Thank you very much for your comments! The differences in DNA sequences across the different genes/region have been commented on in line 261.
Reviewer 2 Report
Dear Editor
In general manuscripts carefully reviewed and hopefully have novelty and new scientific information but author is essential to pay attention to minor revision as I mentioned on the text. There are references missing that needed to be write it down, for example there is no references for morphological identification those fungal isolates?????
So if he/ she is going to accept the our suggestion for minor changes, its will be appreciated.
Kind regards

Author Response
Response to Reviewer 2 Comments
Point 1: Please to be written that where have done your sampling in china , is it North / or south of China??.
Response 1: “southern” has been added following the suggestion.
Point 2: Where did you provide? write it down chemical company......?.
Response 2: The chemical company has been added following the suggestion.
Point 3: Where did you get inoculation method? References??????
Response 3: The reference has been added following the suggestion.
Point 4: You need to show your references as Key identification of Genus and species has been done.
Response 4: The references have been added following the suggestion.
Point 5: You need to make a different row for references belonged, those are accession number from GenBank different than this study.
Response 5: The references have been added following the suggestion.
